# Genome Resequencing for Autotetraploid Rice and Its Closest Relatives Reveals Abundant Variation and High Potential in Rice Breeding

**DOI:** 10.3390/ijms25169012

**Published:** 2024-08-19

**Authors:** Yachun Zhang, Anping Du, Liqi Tong, Gui Yan, Longxiang Lu, Yanni Yin, Xingyue Fu, Huixin Yang, Hui Li, Weizao Huang, Detian Cai, Zhaojian Song, Xianhua Zhang, Yuchi He, Shengbin Tu

**Affiliations:** 1Chengdu Institute of Biology, Chinese Academy of Sciences, Chengdu 610213, China; zhangyachun1225@163.com (Y.Z.); duap@cib.ac.cn (A.D.); llxdewyyx@163.com (L.L.); fuxingyue1@163.com (X.F.); huixin98_yang@163.com (H.Y.); lihui@cib.ac.cn (H.L.); huangwz@cib.ac.cn (W.H.); 2State Key Laboratory of Biocatalysis and Enzyme Engineering, School of Life Sciences, Hubei University, Wuhan 430062, China; tong451027@163.com (L.T.); 202111107010086@stu.hubu.edu.cn (G.Y.); yinyannider@163.com (Y.Y.); dtcai8866@163.com (D.C.); zjsong99@126.com (Z.S.); 20150072@hubu.edu.cn (X.Z.); 3University of Chinese Academy of Sciences, Beijing 100049, China

**Keywords:** rice, tetraploid, tetraploid revertant diploid, genome resequencing, genome variation

## Abstract

Polyploid rice and its reverted diploid show rich phenotypic variation and strong heterosis, showing great breeding value. However, the genomic differences among tetraploids, counterpart common diploids, tetraploid-revertant diploids, and hybrid descendants are unclear. In this work, we bred a new excellent two-line hybrid rice variety, *Y Liang You Duo Hui 14* (HTRM12), using Haitian tetraploid self-reverted diploid (HTRM2). Furthermore, we comparatively analyzed the important agronomic traits and genome-wide variations of those closest relatives, Haitian diploid (HT2), Haitian tetraploid (HT4), HTRM2, and HTRM12 in detail, based on multiple phenotypic investigations, genome resequencing, and bioinformatics analysis. The results of agronomic traits analysis and genome-wide variation analysis of single nucleotide polymorphism (SNP), insertion–deletion (InDel), and copy number variation (CNV) show that HT4 and HTRM2 had abundant phenotypic and genomic variations compared to HT2. HTRM2 can inherit important traits and variations from HT4. This implies that tetraploid self-reverted diploid has high potential in creating excellent breeding materials and in breeding breakthrough hybrid rice varieties. Our study verifies the feasibility that polyploid rice could be used as a mutation carrier for creating variations and provides genomic information, new breeding materials, and a new way of application for tetraploid rice breeding.

## 1. Introduction

Rice, the main food source for half of the world’s population, has a slow growth rate compared with corn. Increasing its yield has become a worldwide goal [1]. Twice “Green Revolutions” in rice breeding, dwarf breeding, and heterosis utilization have significantly increased rice yields [2]. Polyploidy plays an essential role in species domestication, and numerous studies have shown that 70% of angiosperms have had one or more polyploidy events [3,4]. The main cereals, cotton, and oil crops cultivated around the world, such as wheat, cotton, and oilseed rape, have evolved from diploids to allopolyploids [5,6]. From the perspective of systematic evolution, allopolyploids have generated heterozygosity between genomes, increased genome capacity, widened the range of genetic variation, and enhanced tolerance to adverse factors, resulting in a high and stable yield [7,8]. With the combination of the heterologous genome and the occurrence of polyploidy, crop yield has increased significantly. Creatively, scientists have put forward the idea of using the dual advantages of distant hybridization and polyploidy to cultivate new varieties and improve yield in rice [9].

Compared with other crops, rice has the smallest genome, being 40 times smaller than that of wheat and 10 times smaller than that of barley. In addition, the number of chromosomes in rice is also very small [10,11]. From the theory of species ploidy level evolution and chromosome size suitability, it is possible to combine the advantages of polyploidy and genomic interactions to greatly improve (i.e., increase the yield by 50%) or even double the rice yield and obtain new rice varieties with high quality, high yield, and resistance to multiple stressors [9,12,13].

Autotetraploid rice, as a new germplasm, is generated by artificial chromosome duplication of diploid rice, which has attracted extensive attention in rice polyploid breeding in recent years [14]. Compared with diploid rice, autotetraploid rice has some characteristics of other polyploid plants, such as strong plant morphology, large grain size, high 1000-grain weight, increased nutrient content and stress resistance, and higher genetic diversity, which show great potential in rice breeding [9,14,15,16]. However, tetraploid rice generally has low fertility, which affects its yield and further limits the popularization and application [17,18,19,20,21,22]. Currently, some high-fertility tetraploid materials have been developed, such as high-fertility tetraploid rice restorer lines [23], polyploid meiosis stability lines (PMeS) [24], and neo-tetraploid [25,26,27,28,29], which led to an important breakthrough in the utilization of tetraploid rice. Furthermore, the important progress in the construction of three-line hybrid and two-line hybrid tetraploid rice systems for tetraploid heterosis utilization had been made by using traditional breeding and modern biotechnology breeding methods. Tu et al. explored the feasibility of the three-line hybrid tetraploid rice system and produced autotetraploid rice materials fitting in the cytoplasmic male sterility (CMS) system, including male sterile, maintainer, and restoring lines [23]. Afterwards, some important new tetraploid rice materials have been developed for two-line hybrid tetraploid rice systems, such as photoperiod- and thermo-sensitive genic male sterile lines (PTGMS) and thermo-sensitive male sterile lines (TGMS) [30,31].

Although the tetraploid rice heterosis utilization systems were explored and a number of new autotetraploid rice materials had been created, there is no report of commercial rice tetraploid variety developed directly from autotetraploid rice materials. In the genetic process of polyploidization, polyploid rice can revert to diploid rice by self-crossing, which may be caused by uneven chromosome distribution during the formation of male and female gametes in meiosis, and by the anther culture technique, creating rich available materials for breeding [32]. Research has shown that some excellent characteristics of such polyploids may be inherited by diploids. Using rice polyploid recovery to create excellent diploid breeding materials is of great practical significance in improving the genetic characteristics of existing crops and in creating new crop varieties with greater yield potential and better quality [33]. Excellent diploids have been selected from the ploidy-reverted progeny of autopolyploids by self-crossing from extensive test mating groups, and some hybrids have shown strong heterosis, indicating that this method can also provide excellent parent materials for the breeding of hybrid rice with high yield, high quality, and strong resistance [33]. Thus, this suggests a new way for the breeding and utilization of tetraploid rice.

At present, there are some studies on the genomic variation of diploids and tetraploids [21,22,27,28,34,35], but there are almost no studies on the genomic variation among diploids, tetraploids, tetraploid self-reverted diploids, and hybrid descendants of tetraploid self-reverted diploids. A series of analyses and the creation of excellent material for polyploid rice breeding, for example, creating superior revertant mutants using abundant genetic recombination during polyploid rice meiosis, can provide guidance for the selection of new hybrid varieties.

In this study, a new excellent two-line hybrid rice variety, *Y Liang You Duo Hui 14* (HTRM12), was bred using a systematically selected Haitian tetraploid self-reverted diploid (HTRM2) as the male parent. The hybrid variety recently obtained a national certificate. Furthermore, we comparatively analyzed the important agronomic traits and genome-wide variations of Haitian diploid (HT2), Haitian tetraploid (HT4), HTRM2, and HTRM12 in detail, based on multiple phenotypic investigations, genome resequencing technology, and bioinformatics analysis. Our results showed that HT4 and HTRM2 had abundant phenotypic and genomic variations compared to HT2. HTRM2 can inherit important traits and variations from HT4, thus the tetraploid self-reverted diploid has high potential in creating excellent breeding materials and in breeding breakthrough hybrid rice varieties. This study provides insights into parental utilization of tetraploid rice and also provides genomic information and new breeding materials for tetraploid rice breeding.

## 2. Results

### 2.1. Breeding and Field Performance of Superior Hybrid Rice HTRM12

The variety HT2 is a superior restorer diploid line for three-line hybrid rice breeding. HT4 is an autotetraploid rice developed from HT2 by chromosome doubling with colchicine treatment, and HTRM2 is a self-reverted diploid developed from the self-fertile offspring of tetraploid HT4. We crossed excellent rice material *Y58S* and HTRM2 to breed the two-line hybrid rice variety HTRM12. Generally, HTRM12 performed better than the control variety *Feng Liang You 4 Hao* (FLY4) in field trials in the area of the lower and middle reaches of the Yangtze River. HTRM12 had outstanding performance for planting population, plant type, panicle type, and rice grain type (Figure 1A–E) and passed the national variety validation. Although HTRM12 showed a significantly smaller 1000-grain weight than the control variety FLY4 in the yield constitution factors, the yield of HTRM12 was 9.67 tons/ha, which was slightly greater than FLY4 (Figure 1F–I). HTRM12 showed a longer panicle and a significantly smaller plant height than the control variety FLY4 (Figure 1J,K), and its total spikelet number per panicle, seed setting rate, heading date, and whole growth period did not differ significantly from those of FLY4 at different trial sites (Figure 1L–O; Appendix A). This variety had a balanced resistance to biotic and abiotic stress. Its comprehensive rice blast resistance index was 5.2, and it had moderate resistance to rice blast and bacterial blight but was highly susceptible to brown planthoppers based on the national standard test (Table 1). HTRM12 had a stronger tolerance to high temperatures than FLY4 at the pollination stage (Appendix A). This variety had a normal rice quality according to national standards. Its head rice ratio reached 67.8%, and its lowest chalkiness degree was 5.3%. The amylose content of HTRM12 reached 19.5%, and its gel consistency reached 45 mm. The alkali spreading value of HTRM12 was 7-grade, and its smallest grain length-to-width ratio was 3.0 (Appendix A).

### 2.2. Comparison of Agronomic Traits of HT2, HT4, HTRM2, and HTRM12

To determine the differences in agronomic traits and genetic characteristics among HT2, HT4, HTRM2, and HTRM12, the four rice materials were planted together, and several important traits were investigated. Field observations showed that tetraploid HT4 had the fewest branches per panicle, and the number of branches was similar between HT2 and HTRM12. The number of branches in HTRM2 was less than that in HT2 and HTRM12 (Figure 2A). The grains of HT4 had longer awns, and HTRM2 inherited the awn trait from HT4 but had shorter awns (Figure 2B,C). The awn protects seeds from being eaten by animals and can improve the spread of seeds, making it an excellent protective trait [36]. The grain length and grain width of HT4 were significantly higher than those of the diploids, namely HT2, HTRM2, and HTRM12, and the length and width of HT2 and HTRM2 were similar and significantly higher than those of HTRM12 (Figure 2B–E). The tiller number of HT2, HT4, and HTRM12 did not differ much, but the tiller number of HTRM2 was 13, on average, which was significantly higher than that of the other three materials (Figure 2F). However, the plant height of HTRM2 was significantly lower than that of HT4. The plant height of HTRM2 was 11.49% lower than that of HT4, and the plant height of HTRM12 was between that of HT2 and that of HT4, with a significant difference (Figure 2G). Plant height is closely related to photosynthetic efficiency and lodging resistance and directly affects the biomass of rice plants. Therefore, an appropriate reduction in plant height is beneficial for improving the lodging resistance of rice [37]. Although HTRM12 had smaller grains, the number of grains per panicle and the yield per panicle were significantly higher than those of the other three materials; the number of grains per panicle was 4.1 times higher than that of HT4, and the yield per panicle was 3.4 times that of HT4 (Figure 2H,I).

These results indicate that the reverted diploid HTRM2 inherited some excellent traits, such as plant height and awn trait, from tetraploid HT4 and that some traits, such as tiller number, were even better than those of HT2. The number of grains per panicle and yield per panicle of the F_1_ generation of HTRM2 were better than those of its female parent, and it had strong heterosis. The excellent self-reverted diploid selected from progeny of the autopolyploid by self-crossing provided excellent parental material for breeding high-quality, high-resistance hybrid rice, which verified the feasibility that polyploid rice could be used as a mutation carrier for creating variations and breeding excellent varieties of diploid rice.

### 2.3. Genome-Wide Polymorphism Detection in HT2, HT4, HTRM2, and HTRM12

To study the genomic variation differences among HT2, HT4, HTRM2, and HTRM12, whole-genome sequencing was performed using Illumina HiSeq2500, and 23, 24, 20, and 22 Gb of resequencing data, respectively, were obtained. The raw sequence data were deposited in the NCBI Short Read Archive under accession number PRJNA1017988, and the sequencing quality was high (Q20 ≥ 98%, Q30 ≥ 93%) (Appendix A). The GC distribution was normal, and the samples were not contaminated, indicating that the library was constructed and sequenced successfully and could be used for subsequent research (Appendix A).

After quality control and filtering of the sequencing results, there were 133,508,370, 146,785,840, 159,363,942, and 151,883,348 valid reads (total reads) in HT2, HT4, HTRM2, and HTRM12, respectively. The valid reads were then mapped to the *Nipponbare* reference genome. A total of 111,540,478, 121,670,405, 133,024,246, and 125,869,239 reads of HT2, HT4, HTRM2, and HTRM12, respectively, were mapped to the reference genome, and the average coverage depths of HT2, HT4, HTRM2, and HTRM12 were 44×, 48×, 52×, and 50×, respectively (Appendix A). The detailed genome mapping quality and coverage distribution are shown in Appendix A. These results indicate the whole-genome sequencing is high-quality and reliable.

The SNP and InDel variations were examined using the GATK tool (version: 4.1.0.0), with the *Nipponbare* genome as a reference [38]. The total number of SNPs was 3,384,568, 3,438,287, 3,728,757, and 3,475,370, and the total number of InDels was 548,409, 561,615, 608,871, and 569,020 for HT2, HT4, HTRM2, and HTRM12, respectively (Appendix A). The total number of InDels was much less than the total number of SNPs; HTRM2 had the most SNPs and InDels, followed by HTRM12 (Appendix A). The SNPs and InDels were unevenly distributed across chromosomes. Chromosome (chr) 1 had the most SNPs and InDels, and the SNPs and InDels on chr11 and chr12 significantly differed among the four materials (Figure 3A–D and Appendix A). Although HTRM2 was a tetraploid-revertant diploid from HT4, which was derived from HT2 by chromosome doubling, the total SNP and InDel variation in HT4 and HTRM2 were more abundant than that of HT2. This shows that the autotetraploid and its revertant diploid have the potential to produce new variations within new materials.

HTRM12 had the homozygous SNPs with a homozygosity rate of 17.74%, which was slightly lower than the 18.09% homozygosity rate of HT2, and HT4 had the smallest number of homozygous SNPs and the lowest homozygosity rate (Figure 3E,F). Although the total number of InDels was small, the number of homozygous InDels for each material accounted for a large proportion (Figure 3E,G). The homozygous InDel rate was similar between HT2 and HTRM12 (nearly 83%). The homozygous InDel rate of HTRM2 was 75.89% and that of HT4 was only 64.86%. The overall homozygous SNP and InDel variation proportion of tetraploids and revertant diploids was low, and the number of homozygous SNP and InDel variations increased after revertant crosses, providing guidance for breeding practice (Figure 3F,G). Therefore, these results show that HTRM2 and HTRM12 had more abundant variation sites, which is conducive to the cultivation of new materials for breeding superior hybrid rice varieties.

### 2.4. SNP and InDel Site Annotation and Mutation Type Analysis of HT2, HT4, HTRM2, and HTRM12

To explore the potential effects of polymorphic sites on gene function among HT2, HT4, HTRM2, and HTRM12, we performed polymorphic site annotation and mutation type statistical analysis. SNP annotation showed that, after removing the SNP mutations in the upstream and downstream regions, the SNP mutations of the four rice materials occurred in the protein coding region (CDS), accounting for 62%. The intergenic region reached 16% (Appendix A), and missense mutations (34%) accounted for the largest proportion of SNP mutations in the CDS. The proportions of synonymous and stop-gain mutations were slightly different among the four materials. Synonymous mutations reached 27% in HT2 and HTRM2 and 26% in HT4 and HTRM12. Both HT2 and HTRM2 had 1% stop-gain mutations, and both HT4 and HTRM12 had 2% stop-gain mutations (Appendix A). The high proportion of stop-gain mutations, one of the most severe mutation types, in HT4 and HTRM12 indicates that polyploid rice has high mutational ability. In addition, we determined the frequency of SNP mutation types in each sample. Among all samples, the highest mutation frequency was C > T, followed by G > A, T > C, and A > G. The mutation frequency of the other types decreased significantly (Appendix A). Notably, regardless of the mutation type, the mutation frequency was the highest in HTRM2, followed by that in HTRM12.

The InDel annotation results showed that the mutations of HT2, HTRM2, and HTRM12, except for the upstream and downstream regions, accounted for 31% of the mutations in the CDS, while the mutations in HT4 accounted for 30%. Among the mutation types in the CDS of HT2, HT4, HTRM2, and HTRM12, infra-shift mutations accounted for 10%, and frameshift mutations accounted for 21%. The frameshift mutations in HT4 accounted for 20%. The InDel mutations in the untranslated region (UTR) and the stop-loss type in the CDS of HT4 accounted for 22% and 49.0%, respectively, which were the highest among the four samples. This indicates that polyploid rice HT4 has high potential for creating new rice germplasm (Appendix A).

The effects of SNP and InDel variants on gene function were predicted using the SnpEff tool (version: 4.4) [39], and four types with high, moderate, low, and modifier effects were clustered (Appendix A). The number of effects of SNP and InDel mutations on genes followed the order Modifier > Moderate > Low > High and Modifier > High > Moderate > Low, respectively. In the four materials, HTRM2 and HTRM12 had a higher number of high and moderate SNP and InDel variation types, indicating a greater influence on gene function and the corresponding phenotype than low and modifier types.

In summary, tetraploid HT4 had more variations (SNPs and InDels) in important genome regions (such as CDS), and tetraploid-revertant diploid HTRM2 as well as hybrid descendant HTRM12 harbored more high and moderate variation types. This indicates that tetraploid can be used as a mutation carrier for creating novel diploid materials by ploidy reverting mutation, and the revertant diploid with rich variations has the potential to breed excellent hybrid rice.

### 2.5. GO Terms and KEGG Enrichment Analysis of SNP Mutant Genes of HT2, HT4, HTRM2, and HTRM12

To further investigate the causes of the differences in HT2, HT4, HTRM2, and HTRM12, we performed Gene Ontology (GO) and Kyoto Encyclopedia of Genes and Genomes (KEGG) enrichment analysis for the genes in which homozygous SNP and homozygous InDel mutations occurred.

Items enriched in HT2 and HTRM2 were mostly the same, such as “L-arabinose metabolic process”, “reproduction”, “sulfur compound transport”, and “dolichyl monophosphate biosynthesis process” (Figure 4A,C). Items enriched in HT4 were similar to those enriched in HT2 and HTRM2, but there were many unique entries, such as “defense response to oomycetes”, “negative regulation of endopeptidase activity”, and “protein insertion into membrane” (Figure 4B). Endopeptidases play a key role in the mobilization of storage proteins during seed germination [40]. Endopeptidases can be involved in biotic stress and pathogen defense [41]. Polyploid rice is more resistant to stress, and endopeptidases likely play a role in the stress response of polyploid rice. The enriched GO terms in HTRM12, such as “meiosis II”, “mRNA splicing”, “leucine metabolic processes”, “aging”, and “positive regulation of protein serine/threonine kinase”, were significantly different from those in the other three samples; these entries existed only in HTRM12 (Figure 4D). These items all play an important role in plant development. For example, mRNA splicing is a very important biological process in eukaryotic gene expression. Through RNA splicing, many functional mRNAs with coding information can be produced, which is vital to the development and evolution of organisms [42]. Meiosis is the cytological basis of heredity and variation, and it plays a key role in the life process of plants [43].

The KEGG enrichment results showed the same change trend as the GO term enrichment analysis among the four materials. In the results, HT2 and HTRM2 were similar. The pathway with the highest gene percentages in HT2 and HTRM2 was “carbon metabolism”, followed by “RNA transport” and “purine metabolism”. Interestingly, there were a small number of unique pathways between HT2 and HTRM2. For example, “mRNA surveillance pathway” was unique to HT2, and “valine, leucine, and isoleucine biosynthesis” and “peroxisome” were unique to HTRM2. These KEGG pathways may cause differences between HT2 and HTRM2 in RNA, protein metabolism, and oxidative stress response, although HTRM2 was a tetraploid-revertant diploid derived from HT4 by chromosome doubling of HT2 (Figure 5A,C). The enrichment results of HT4, HT2, and HTRM2 were similar, and the pathway with the largest number of genes was “RNA transport”, followed by “purine metabolism” and “tryptophan metabolism” (Figure 5B). The above-mentioned pathways are all primary metabolic pathways [44]. In contrast, the pathway with the most enriched genes for HTRM12 was “phenylpropane biosynthesis”, followed by “peroxisome” and “phagosome” (Figure 5D). Phenylpropane metabolism is one of the most important plant secondary metabolic pathways, producing more than 8000 metabolites, which play an important role in plant growth and development and plant–environment interactions [45]. The phenylpropanoid metabolites play a vital role in many aspects, which can regulate plant growth and development and play a role in disease resistance and anti-aging [46]. Unique KEGG pathways in HTRM12, such as “phenylpropanoid biosynthesis”, “ribosome biogenesis in eukaryotes”, “sulfur metabolism”, and “photosynthesis”, may confer the hybrid rice variety HTRM12 with the highest grain number per panicle and weight per panicle among the four materials.

### 2.6. GO Terms and KEGG Enrichment Analysis of InDel Mutant Genes of HT2, HT4, HTRM2, and HTRM12

The enrichment results of GO terms of InDel mutant genes showed that the items with the most genes in HT2, HT4, and HTRM2 were related to stress and environmental responses. For example, genes related to “response to biological stimulation” and “regulation of response to stimulation” accounted for the largest proportion in HT2 and HT4 (Figure 6A,B). The items with the most genes in HTRM2 were “response to osmotic stress” and “regulation of response to water deprivation” (Figure 6C). There were no stress response items enriched in HTRM12, and it differed from the other three samples, except for the enrichment of “P-body assembly”, which was also found in HT2 (Figure 6D). Other high-volume items such as “response to inorganic substances”, “positive regulation of cellular component biosynthesis”, “RNA phosphodiester bond hydrolysis”, “sphingolipid metabolic process”, and “basic amino acid transmembrane transport” were only present in HTRM12 (Figure 6D). These results showed that the gene similarity of the InDel homozygous mutation between HT2 and HT4 was high, while that in HTRM12 was relatively unique. The gene function of its InDel homozygous mutations differed from those of the other three materials.

The KEGG enrichment analysis results of the four samples were almost the same, except for a slight difference in the number of genes. For example, the most enriched genes in metabolic processes were related to “global and overview maps”, followed by some primary metabolic processes, such as “carbohydrate metabolism”, “amino acid metabolism”, and “lipid metabolism”. Most genes enriched in the gene information process were related to “translation”, followed by “folding, sorting, and degradation” and “transcription”. The same was true for entries in other metabolic processes (Appendix A). The InDel mutant genes caused differences in GO term enrichment results among the four samples, especially between HTRM12 and the other three samples, while the KEGG enrichment results were the same among HT2, HT4, HTRM2, and HTRM12.

There were 47,351, 41,670, 48,358, and 48,209 genes harboring at least one homozygous InDel (homo-InDel gene) in HT2, HT4, HTRM2, and HTRM12, respectively (Figure 7A). There were 703, 652, 940, and 1082 unique homo-InDel genes (only present in one material) in HT2, HT4, HTRM2, and HTRM12, respectively (Figure 7A,B). A genome-wide association study (GWAS) phenotype enrichment analysis [47] was performed to identify the candidate genes for special phenotypes in each material. Two terms, “alkali spreading value” and “flag leaf length”, were only found in HT2. “Awn presence” was found only in HT4. Four terms, i.e., “amylose content”, “awn presence”, “pericarp color”, and “flowering time at Arberdeen”, were found in HTRM2, and three terms, i.e., “heading data”, “shattering degree”, and “tiller number”, were found in HTRM12. Interestingly, “awn presence” was present in both HT4 and HTRM2, which agreed with their awn phenotypes (Appendix A).

We focused on the genes responsible for the grain and awn phenotypes. The GWAS phenotype enrichment analysis showed that LOC_Os12g24080 (*large2*) was enriched in HT4 (Appendix A). The *Large2* gene has been reported to regulate the rice grain type, and the loss of function of this gene leads to a larger grain type [48]. Moreover, GWAS association analysis has previously shown that this gene is associated with the awn phenotype [49]. The phenotypes of HT4, such as the large grain and awn phenotypes, were obviously different from those of the other three materials, which is likely related to the variations in the *Large2* gene. Analysis of homo-InDel variation showed that an adenine base was inserted in the 2nd intron of this gene, and G and A bases were absent at the splicing of the 6th exon and 6th intron (Figure 7C). These variations may lead to a change in the pre-mRNA splicing of the *Large2* gene, thus causing a change in the function of the translated protein. These variations are likely to be important to the grain and awn phenotype differences in HT4 compared with the other three materials. Interestingly, HTRM2 also showed an awn phenotype, but the GWAS phenotype enrichment analysis of its homo-InDel genes highlighted a new gene, LOC_Os06g44930 (Appendix A), which has also been associated with the awn phenotype of rice using GWAS [47]. Analysis of the homo-InDel variation in the coding region of this gene showed that there were two homo-InDel variations in the coding region, leading to changes in the coding protein sequence (Figure 7D). Collectively, the identified genes and corresponding InDel variations, for example, the awn genes, may be important for special phenotype variations in the four materials, which may have potential in polyploidy rice breeding.

### 2.7. Identification and Comparison of Copy Number Variation (CNV) for HT4, HTRM2, and HTRM12

Copy number variation (CNV), which is expressed as an increase (gain) or decrease (loss) in the copy number of genomic segments, is widely distributed in the human genome and is an important genetic basis for individual differences [50]. CNVs may be biologically important for species-specific genome composition, species evolution, phylogeny, and the expression and regulation of genes in specific regions of the genome [51]. Therefore, CNVs have become a topic of interest in genomics research in recent years. To study CNVs in rice with ploidy changes, the HT4, HTRM2, and HTRM12 CNV segments were identified using the HT2 genome as a control. There were no CNV segments identified on chr3, chr5, and chr6, and CNV segments were identified for the three samples on chr1, chr2, chr4, chr7, and chr8 (Figure 8A). However, only HT4 and HTRM12 contained CNVs on chr9 and chr11, and only HTRM2 and HTRM12 contained CNVs on chr10 and chr12 (Figure 8A). The HT4 genome had a CNV interval of 13.20 Mb, accounting for 3.14% of the whole genome relative to the HT2 genome. The HTRM2 genome had a CNV interval of 16.06 Mb, accounting for 3.82% of the whole genome relative to the HT2 genome. Compared with the HT2 genome, there was a 22.10 Mb CNV interval in the HTRM12 genome, accounting for 5.26% of the whole genome (Figure 8B). Analysis of the CNV distribution on different chromosomes of the samples revealed that most of the deletions of the copy number regions (CNRs) and copy number segments (CNSs) occurred in three samples, especially HT4 and HTRM2, where only CNSs were lost (Figure 8C,E). However, there were CNR gains on chr12 of HTRM12 (Figure 8G). According to the principle of CNV calling, most lost CNVs have a small copy ratio [52]. There were more CNSs with extremely small copy ratios of loss CNVs on chr4, chr7, and chr11 in HT4 (Figure 8D). In HTRM2, the lost CNVs with extremely small copy ratios were distributed on chr1 and chr10 (Figure 8F). In HTRM12, they were distributed on chr1, chr10, and chr11 (Figure 8H). These different chromosome distributions of lost CNVs with extremely small copy ratios in each material may contribute different characteristics to each rice variety.

Analysis of the genes presented in the CNV segments of the three samples revealed that fewer genes were present in the CNV segments of HT4 than in the CNV segments of HTRM2 and HTRM12, but the percentage of CNV-specific genes presented only in HT4 was the highest (59.87%) (Figure 9A–C). The CNV segments of HTRM12 contained the most genes (1452 genes) but the lowest percentage of CNV-specific genes (39.81%) (Figure 9C). We also explored the GO term enrichment analysis of specific genes contained in the CNSs of HT4, HTRM2, and HTRM12. The top five items in the biological processes (BPs) in HT4 were all metabolic processes, including primary and secondary metabolism. The most frequent items in the cellular components (CCs) were extracellular regions and cell membranes, and the top five entries in the molecular functions (MFs) were all related to binding, including anion binding, nucleotide binding, and small molecule binding (Figure 9D). The genes in the specific CNV compartment in HTRM2 were enriched in more BP items. In addition to metabolic processes, there were catabolic processes. In contrast to HT4, the most enriched items in the CCs of HTRM2 were enzyme complexes, catalytic complexes, and protein complexes. However, the top five items in the MFs of HTRM2 were similar to those of HT4 in that they were all binding-related items (Figure 9E). The genes in the specific CNV compartment in HTRM12 were enriched for more entries in BPs and were different from those in HT4 and HTRM2, which were mainly enriched in signaling-related items (Figure 9F). The entries enriched in the CCs of HTRM12 were more similar to HT4 in that they were membrane-related items. The top five enriched items in the MFs of HTRM12 were related to kinase activity and binding (Figure 9F). These specific items of gene enrichment for specific CNV intervals may be candidates for phenotypic differences among the three materials.

## 3. Discussion

At present, breeders are promoting the application of tetraploid rice by cultivating highly fertile tetraploid materials and constructing tetraploid heterosis utilization systems and have made continued progress (see Section 1) [23,24,25,26,27,28,29,30,31]. However, as far as we know, there is no commercial variety directly bred with tetraploid. In this study, we used tetraploid rice HT4 as a mutant carrier to breed a new excellent diploid material HTRM2, and then bred the hybrid rice variety HTRM12. This hybrid rice variety has passed the national variety approval. It has excellent agronomic traits, with a high yield of 9.67 tons/ha, high quality, and multi-resistance (Figure 1; Appendix A). This hybrid rice is suitable for planting in the middle and lower reaches of the Yangtze River, China. Furthermore, the agronomic traits analysis of four materials with ploidy change shows the tetraploid self-reverted diploid HTRM2 can inherit some excellent traits of tetraploid HT4, and some traits are even better than original diploid HT2 (Figure 2).

Diploid can be generated with low probability from tetraploid self-bred progeny in rice. The reverted diploid rice may have more phenotypic variation than the original diploid. Xing et al. found the diploid can be produced in tetraploid self-bred progeny in rice [53]. Xiong et al. generated the self-reverted diploids from self-bred progenies from japonica tetraploids, indica tetraploids, and indica-japonica hybrid tetraploids, respectively. The selfing-reverted diploids showed great phenotypic variations and strong heterosis [33]. This reverted diploid can also be generated by in vitro gametocyte culture of tetraploid. For example, Li et al. generated reverted diploid rice by anther culture of PMeS, 9311-4x, and other tetraploid rice varieties, and these reverted diploid rice varieties showed high quality [54].

In summary, this study verified that tetraploid rice can be used as a mutation carrier to breed excellent parent rice materials and can cultivate excellent hybrid rice varieties with high yield, high quality, and high resistance. Thus, it provides excellent new germplasm for rice breeding and a new way for the application of tetraploid rice in breeding.

Polyploidization can cause genetic redundancy when the homologous or heterologous genomes are combined in the course of the evolution of plants, which will lead to the “genome impact”. In order to cope with the “genome impact” caused by the genome combination, the new polyploid will establish a new balance of the genome through chromosome rearrangement, sequence elimination, gene variation, and so on. On the other hand, plants will also undergo polyploidization to diploidization during the process of evolution. This will induce genome loss, variation, and gene expression changes. This “polyploid cycle” is considered to promote plant diversity and adaptive evolution [55,56]. In this study, we generated the tetraploid rice HT4 by artificial chromosome duplication of diploid rice HT2 and selected the excellent diploid HTRM2 from tetraploid self-bred progeny diploid. Hence, we constructed an artificial “polyploid cycle” in rice.

In order to study the relationship between genomic variation and ploidy change, the genomic variation of four materials was analyzed. Through genome-wide SNP and InDel variation analysis, we found that the number of variations in four materials conforms to the following rules: HTRM2 > HTRM12 > HT4 > HT2 (Figure 3E; Appendix A). The tetraploid self-reverted diploid HTRM2 has the largest number of SNP and InDel variants, and the variation in HTRM2 and HTRM12 is more serious because of the more high, moderate SNP and InDel variation types discovered in the two materials. Specially, the stop-gain variation type of SNP is higher in HT4 and HTRM12, and the percent of InDel variation in the untranslated region (UTR) and the stop-loss type in the CDS of HT4 is highest among four materials. These results suggest that the genomes of rice subjected to the artificial “polyploid cycle” and hybridization are very different from those of the original diploid. At present, a small number of studies have reported the genomic variation between the original diploid and the corresponding tetraploid. Fozia et al. sequenced the genomes of diploid and corresponding tetraploid in rice for studying the embryonic sac fertility in autotetraploid. A large number of SNPs, InDels, SVs, and CNVs were detected between diploids and corresponding tetraploids. It was speculated that changes in the number of chromosomes may induce changes in the genome sequence and produce specific variations in tetraploids [22]. In our study, many variations were detected after chromosome ploidy changes in rice, which further confirms this speculation. Thus, our work provides a good method by using an artificial “polyploid cycle” for creating variation in rice. Moreover, it provides genomic resources for genetic research in the “polyploid cycle” process and for molecular breeding of polyploid and corresponding diploid rice.

GO and KEGG enrichment entries of homozygous SNP were similar among HT2, HT4, and HTRM2, and HTRM12 mostly differed from them. Although the GO entries of HT4 are similar to those of HT2 and HTRM2, there are some stress-related entries specifically enriched in HT4, such as “defense response to oomycetes”, “negative regulation of endopeptidase activity”, “protein insertion into membrane”, etc. These involved genes may contribute to the HT4 higher resistance. Interestingly, there were many different entries related to plant development, such as “meiosis II”, “mRNA splicing”, “leucine metabolic processes”, etc., that were significantly enriched in hybrid HTRM12. These involved genes may contribute to HTRM12’s excellent agronomic properties.

GO enrichment analysis of InDel homologous variation genes found that the stress and environmental response-related entries were enriched in HT2, HT4, and HTRM2, such as “response to biological stimulation”, “regulation of response to stimulation”, and so on. This indicates that stress-related genes were inherited in different ploidy materials. Interestingly, the above entries were not enriched in HTRM12. But some special entries, such as “response to inorganic substances” and “positive regulation of cellular component biosynthesis”, were only present in HTRM12. It may be because HTRM12 is a hybrid rice and the genes embraced in the above entries are heterozygous. However, KEGG enrichment entries of genes with homozygous InDel variants were similar in the four materials. Interestingly, the analysis of the homo-InDel gene found that most of the homo-InDel genes were shared by the four materials, indicating that this gene was inherited in the four materials. The specific homo-InDel gene analysis found that two different awn genes were enriched in HT4 and HT2, respectively, indicating that the InDel variations occurred in the two different awn genes during ploidy changes.

CNVs are reported to impact the characteristics and environmental adaptation in plants and animals [57,58]. The identification and analysis of CNV intervals for HT4, HTRM2, and HTRM12 were consistent with the SNP and InDel results. The size of CNV intervals followed the trend HTRM12 > HTRM2 > HT4, and the size ranking of specific CNV intervals was the same (Figure 8). Compared to the whole genome, HT4, HTRM2, and HTRM12 had 3.14%, 3.82%, and 5.62% CNVs (Figure 8B), respectively, with some specific CNV intervals containing 543, 493, and 578 genes, respectively (Figure 9A–C). GO term enrichment analysis of the genes at specific intervals of the three materials showed that HTRM12 was significantly different from HT4 and HTRM2 (Figure 9D–F). These genes with specific intervals may have resulted in the outstanding morphological characteristics of HTRM12 in the variety test trial.

In summary, our results indicate that genomic variation can occur in the process of chromosome ploidy change in rice, and some variations can be inherited to the reverted diploid from tetraploid rice. The reverted diploid rice has a large variation relative to the original diploid rice, so it can be used to breed high-yield and strong heterosis rice materials. Therefore, this study provides genomic information, new breeding materials, and a new way of application for tetraploid rice breeding.

## 4. Materials and Methods

### 4.1. Rice Material and Phenotype Examination

The variety *Haitian* (HT2) is a superior diploid restorer line for three-line hybrid rice breeding. HT4 is an autotetraploid rice developed from HT2 by chromosome doubling with colchicine treatment [59], and HTRM2 is a high yield, high quality, and high combining ability line that has been selected from the self-bred progeny of tetraploid HT4 for many years. We crossed the excellent two-line sterile rice material *Y58S* and HTRM2 to breed the two-line hybrid rice variety *Y Liang You Duo Hui 14* (HTRM12) in the 2014 year.

For Figure 1 and Appendix A, the data were collected from results of the variety test in 18 regional trial sites distributed in Anhui, Fujian, Henan, Hubei, Hunan, Jiangsu, Jiangxi, and Zhejiang provinces of the area of the lower and middle reaches of the Yangtze River, China. The experiment was carried out in accordance with the Agricultural Industry Standard of the People’s Republic of China [60]. The varieties were sown and transplanted at the same time, the fertilization level was medium to upper, and cultivation management measures were the same as local field production. The values in bar graphs were shown as means ± SD (*n* = 18 trial sites). For each trial site, the materials were planted on an area of 13 square meters in three replicates (three plots), and the average value of the three repeats of each agronomic trait represents the observed value of corresponding traits. The morphological trait measurement methods are described in detail as follows: Yield: Harvest according to the varieties after maturity in separate plots, drying, weighing after cleaning, calculating the yield per hectare. Number of Effective Panicles: At each trial site, the average panicle number of continuous 10 plants in the middle region of two repeat plots was investigated (a panicle with less than 5 grains is not considered an effective panicle) at the rice maturation stage and then converted to ten thousand/ha, which was used as the effective panicle number of the trial site. Total Grain Number Per Panicle: The sum of the number of grains and the number of falling grains with more than 1/3 of the filling degree of 5 plants divided by the total number of panicles of 5 plants. 1000-Grains Weight: The weight of 1000 dry solid grains; each variety is randomly selected as two 1000 grains, respectively weighed; the difference is not greater than 3% of its average value; take two repeated average values. Panicle Length: The length from the node to the top of the panicle (without awns) is taken as the average of all the rice panicles in 5 plants after rice maturity. Plant Height: The length from the top of the panicle (without awns) to the base of the stem was measured at 10 plants in the middle of the plot at the mature stage of rice, and the average value was taken as the plant height of the trial site. Total Spikelet Number Per Panicle: Total spikelet number of 5 plants divided by total panicle number of 5 plants. Seed Setting Rate: Total grain number per panicle divided by total spikelet number per panicle. Heading Date: The heading period is from the sowing time to the day when 80% of the panicles in the plot are exposed to the blade sheath; take the average value of 3 plots as the heading date of each variety in each trial site. Whole Growth Period: The number of days from the sowing date to the maturity date; take the average value of 3 plots as the whole growth period of each variety in each trial site.

To compare the agronomic properties in Figure 2, the four rice materials HT2, HT4, HTRM2, and HTRM12 were grown under normal paddy conditions from November in the winter of 2021 to April in the next year in Sanya City, Hainan, China. After seedling cultivation, each material was transplanted into a plot, respectively, and 30 plants were planted in each plot at a distance of 16 cm × 26 cm. The field management measures were consistent with the local field production methods. The morphological trait measurements were as follows: Plant height: At the rice mature stage, take about 30 individual plants in the middle of the plot, measure the length from the top of the panicle (without awn) to the stem base, and take the average value as the plant height of the material. Tillering number: At the rice maturity stage, take about 30 individual plants in the middle of each plot and count the tillering number of each plant, and then take the average value as the number of tillerings of the material. Grain number per panicle: For each material, at the maturity stage, harvest about 30 main panicles (the first heading panicles) of a single plant in the middle of the plot, count the number of solid grains per panicle of each material (more than 1/3 filling degree), and take the average as the number of grains per panicle of the material. And a typical panicle of each material was photographed (Figure 2A). Grain weight per panicle: After the main panicle was dried at 50 °C for 5 days, the grain was separated from the panicle, and the weight of each single panicle was weighed after the blighted grain was removed, and the average value was taken as the grain weight per panicle of the material. Grain length and grain width: After mixing all single panicle grains of each material, respectively, take about 300–500 grains and measure the grain length and grain width using an automatic seed analyzer (Wan Sheng SC-G, Hangzhou Wanshen testing Technology Co., Ltd., Hangzhou, China), and 10 representative grains of each material were photographed.

### 4.2. The Disease and Pest Resistance Test

For Table 1, the rice blast resistance test data were obtained from the results of 6 trial sites (Zhejiang, Hunan, Hubei, Anhui, Fujian, and Jiangxi province, China) of the rice variety regional test in the 2019 year, and the data in the table represent the average values of the 6 test sites. The identification of each test site was carried out in accordance with the Agricultural Industry Standard of the People’s Republic of China [61]. The susceptibility index of rice blast = Disease grade of leaf blast of seedling × 25% + Disease grade of panicle blast × 25% + Disease grade of the spike loss × 50%; Here, the evaluation standard of disease grade of leaf blast of seedling is: 0 grade (incidence rate = 0), 1 grade (incidence rate ≤ 5.0%), 3 grade (incidence rate = 5.1~10.0%), 5 grade (incidence rate = 10.1~25.0%), 7 grade (incidence rate = 25.1~50.0%), 9 grade (incidence rate ≥ 50.1%); the standard of disease grade of the spike loss is: 0 grade (loss rate = 0), 1 grade (loss rate ≤ 5.0%), 3 grade (loss rate = 5.1~15.0%), 5 grade (loss rate = 15.1~30.0%), 7 grade (loss rate = 30.1~50.0%), 9 grade (loss rate ≥ 50.1%); the standard of disease grade of the panicle blast is: 0 grade (rate of infected panicle = 0), 1 grade (rate of infected panicle ≤ 5.0%), 3 grade (rate of infected panicle ≥ 5.1% and ≤10.0%), 5 grade (rate of infected panicle ≥ 10.1% and ≤25.0%), 7 grade (rate of infected panicle ≥ 25.1% and ≤50.0%), 9 grade (rate of infected panicle ≥ 50.1%).

The test of resistance to white leaf blight was carried out in Anren, Hunan Province, in 2019 and 2020, respectively. The test method was carried out according to the standard of technical regulations for bacterial blight resistance of rice varieties in Hunan Province [62], and the data in the table were the average values of two years’ data. The evaluation standard of disease grade is: 0 grade (lesion length = 0 cm), 1 grade (some small lesion around incision and lesion length ≤ 2 cm), 3 grade (lesion length > 2 cm and ≤1/4 of leaf length), 5 grade (lesion length > 1/4 and <1/2 of leaf length), 7 grade (lesion length > 1/2 and <3/4 of leaf length), 9 grade (lesion length > 3/4 of leaf length).

The test of resistance to rice planthopper was performed at the seedling stage, according to the international standard seedling group screening method (SSST method), with 3 replicates. The data in Table 1 are the average of the results of 2019 and 2020. For the brown planthopper resistance test, the evaluation standard is: 0 grade (no suffering from pests), 1 grade (suffering from pests very slightly), 3 grade (partial yellowing of the first and second leaves of most plants), 5 grade (obviously yellowing, the plants wither or die of about half the plants), 7 grade (more than half the plants died), and 9 grade (all plants died).

### 4.3. The High Temperature Resistance Test

For Appendix A, the high temperature resistance test in field condition was carried out under the natural high temperature conditions from mid-July to mid-August in Wuhan, Hubei Province. HTRM12 and control FLY4 were sown at the same time in three periods. The tillering that continuously lasted more than 5 days under average daily temperature above 32 °C and the highest temperature above 35 °C was taken as the tillering that had been subjected to high temperature stress after heading period, and the average seed setting rate was taken as the seed setting rate under high temperature stress in field conditions. The tillering with the average seed setting rate, which did not experience the above high temperature condition after heading period, was taken as the seed setting rate under normal temperature conditions. Relative heat resistance coefficient = Seed setting rate at high temperature condition of testing variety/Seed setting rate at high temperature condition of control variety.

The high-temperature test of variety in pot culture was carried out in a glass greenhouse. The two rice varieties were planted in pots, setting 4 replicates. When the first spikelet of each material was flowering, the continuous 5 days of high temperature treatment in a glass greenhouse was applied (the air temperature in a glass greenhouse is about 40 °C), and then the materials were transferred to a natural condition (average daily air temperature of about 32 °C during the flowering period) for growth to maturity. At the same time, normal temperature control was set under natural conditions (the average daily temperature of the flowering period was about 32 °C), and then the seed setting rate of high temperature treatment and normal temperature control was tested. The relative heat resistance coefficient is calculated in the same manner as above. The combined heat resistance coefficient is the average value of the relative heat resistance coefficient of the two conditions.

### 4.4. Genome Sequencing and Data Quality Assessment

The construction of the library for genome sequencing and sequencing was performed by the Shanghai Ouyi Biomedical Technology Company (Shanghai, China). Briefly, DNA was extracted from the young roots of HT2, HT4, HTRM2, and HTRM12 using the hexadecyl trimethyl ammonium bromide (CTAB) method. Moreover, 1000 ng of DNA in 52.5 μL was sheared to 350 bp–500 bp using the Covaris S220 (Covaris, LLC, Woburn, MA, USA) at 175 W, 5% duty, 200 cycles for a duration of 20 s. The TruSeq Nano DNA LT Sample Prepararion Kit (Illumina, Inc., San Diego, CA, USA) was used in library construction following the manufacturer’s instructions. After the library was qualified, whole-genome sequencing was performed by double-end sequencing using the Illumina HiSeq2500 sequencing platform (Illumina, Inc., San Diego, CA, USA). After the raw sequencing data (raw data) were obtained from sequencing, the adaptor sequences, reads with a number of N ≥ 5, and sequences with an average base quality value of less than 20 were removed to obtain clean reads using the software FASTP (version: 0.23.4) [63]. The results of sequencing data quality control can be found in Appendix A.

After removing the reads with a length less than 75 bp and an average base quality value less than 15, the clean reads were aligned to the reference genome (*Nipponbare*, MSU-release7) using BWA (version: 0.7.12) [64], with the “bwa men” alignment algorithm and default parameter. Then, the format conversion and ranking of alignment results were performed using SAMtools (version: 1.4) [65], and the PCR repeat reads were removed using Picard (version: 4.1.0.0) (https://broadinstitute.github.io/picard/, accessed on 10 October 2020). The final results were analyzed by Qualimap 2 software (version: 2.2.2-dev) [66] for quality control. The results of mapping clean reads to the reference can be found in Appendix A.

### 4.5. Analysis of Variation Information

The results of alignment to the reference genome (*Nipponbare*, MSU-release7) (BAM files) were analyzed for SNP and InDel variation information. Briefly, SNP and InDel detection were performed using the Haplotypecaller module of GATK4 software (version: 4.1.0.0) [38]. The criterion of variation quality value (QD) ≥ 2.0 was chosen for filtering the real SNP and InDel variations, in which the QD is equal to the variant quality value (Quality) divided by the depth of coverage (Depth). In fact, the QD reflects the variant quality value per unit depth, as most of the false-positive variants had a QD less than 2. The detected SNP and InDel were annotated with SnpEff software (version: 4.4) [39]. The detected genomic variation information was visualized using Circos plots [67] (Figure 3A–D).

### 4.6. Gene Ontology (GO) and Kyoto Encyclopedia of Genes and Genomes (KEGG) Enrichment Analyses

Homozygous SNP and InDel variant genes within the GO database were annotated, and the number of differential genes in each GO entry was calculated. Statistical analysis was performed using hypergeometric tests. The annotation results were compared with the genomic background to screen GO entries that were significantly enriched in the variant genes. The calculated *p*-value was corrected through multiple hypothesis tests and then used as a threshold for GO entries with significantly enriched differentially expressed genes with a q-value < 0.05. GO annotation results were visualized using Revigo [68]. Statistical analysis of genes with homozygous SNP and InDel specificity was conducted to determine the enriched KEGG pathways. The statistical principles and methods were similar to GO analysis, and significantly enriched pathways were identified through analysis. The KEGG annotation results were visualized using R language [69].

### 4.7. GWAS Phenotype Enrichment Analysis

The GWAS phenotype enrichment analysis in comprehensive annotation of rice multi-omics data (CARMO), an integrated annotation platform for functional exploration of rice multi-omics data, is a method of gene functional enrichment analysis like GO enrichment and KEGG enrichment analysis, but the raw gene set (reference gene set) of GWAS phenotype enrichment analysis was collected from genes affected by a set of single nucleotide polymorphisms (SNPs) associated with a certain trait identified from genome-wide association studies (GWAS) and integrated in CARMO. For any input study or gene list, it will be apparent which genes participate in regulation of key agronomic traits [47]. In this study, the homo-InDel genes of HT2, HT4, HTRM2, and HTRM12 were used for Venn diagram analysis to identify the special homo-InDel genes in the four materials (Figure 7A,B). The special homo-InDel genes of each material were used for GWAS phenotype enrichment analysis by the web-based tool CARMO, and the output-enriched gene lists of each material were compiled into Appendix A.

### 4.8. Identification and Comparative Analysis of Copy Number Variation Intervals

Copy number variation is one of the major types of polymorphisms in rice genome research. In this study, to identify the CNV intervals of the test material relative to the HT2 reference genome, CNVkit software (version: 0.9.7) [52] was used to detect CNVs in our samples. In general, the whole genome of the test sample was divided into small windows of 1 Mb in the analysis (CNV region), and the ratio of read coverage of the test sample to the reference sample and the log_2_
^ratio^ values were calculated to identify chromosomal segment gain or loss using the sliding window algorithm. The log_2_
^ratio^ < −0.4 or log_2_
^ratio^ > 0.3 was used as an indication of chromosomal segment loss or gain, respectively.

### 4.9. Statistical Methods

The data in Figure 1 were analyzed and plotted using GraphPad Prism (version: 9.0.0) software (Boston, MA, USA). The normality of data was tested (Anderson–Darling test and D’Agostino and Pearson test), and Student’s *t*-test (two-tailed) was used for difference significance analysis. Specially for Figure 1K, the Welch’s *t*-test (two-tailed) was used for difference significance analysis because the variances (SDs) between two samples are not equal. For Figure 1H–J, as the data of FLY4 does not fit a normality, the Mann–Whitney U test (two-tailed) was used for difference significance analysis. ns presents no significant difference; *, *p* < 0.05; **, *p* < 0.01; ***, *p* < 0.001; ****, *p* < 0.0001.

In Figure 2, data analysis was performed using R language, and Student’s *t*-test (two-tailed) was used for difference significance analysis after normality of the data was tested (Anderson–Darling test and D’Agostino and Pearson test). Moreover, ns presents no significance; *, *p* < 0.05; **, *p* < 0.01.

## Figures and Tables

**Figure 1 ijms-25-09012-f001:**
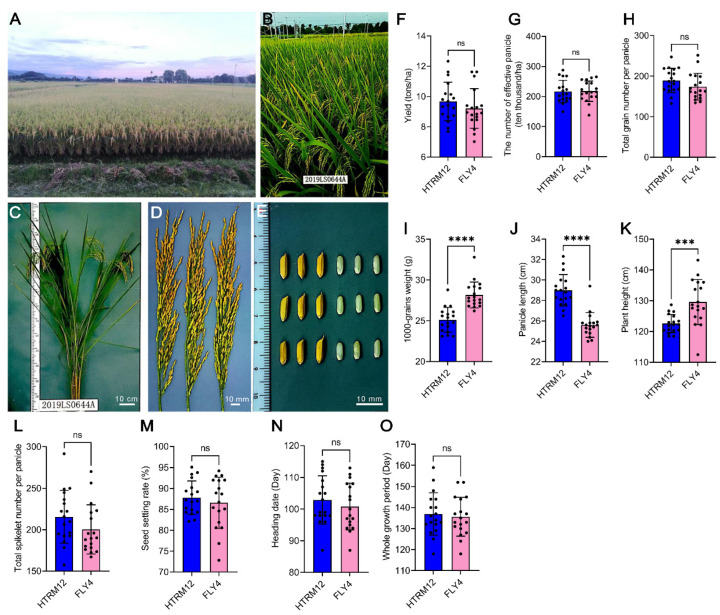
The field performance of hybrid rice HTRM12. (**A**,**B**) the field performance of HTRM12 population in production trial at maturity stage (**A**) and grain filling stage (**B**); (**C**) the plant phenotype of HTRM12 at grain filling stage; (**D**) the panicles phenotype of HTRM12 at grain filling stage; (**E**) the caryopses and brown grains phenotype of HTRM12 at maturity stage; (**F**–**I**) comparison of the yield and yield constitution factors between HTRM12 and control variety FLY4; (**J**) comparison of the panicle length between HTRM12 and control variety FLY4; (**K**) comparison of the plant height between HTRM12 and control variety FLY4; (**L**–**O**) comparison of total spikelet number per panicle, seed setting rate, heading date, and whole growth period between HTRM12 and control variety FLY4. Data collected from different regional trial sites were used for plotting, and values were shown as means ± SD (*n* = 18 trial sites). Student’s *t*-test was used for difference significance analysis (two-tailed). Specially for (**K**), the Welch’s *t*-test (two-tailed) was used for difference significance analysis. For (**H**–**J**), the Mann–Whiney U test (two-tailed) was used for difference significance analysis. ns presents no significant difference; *** presents *p* < 0.001; **** presents *p* < 0.0001.

**Figure 2 ijms-25-09012-f002:**
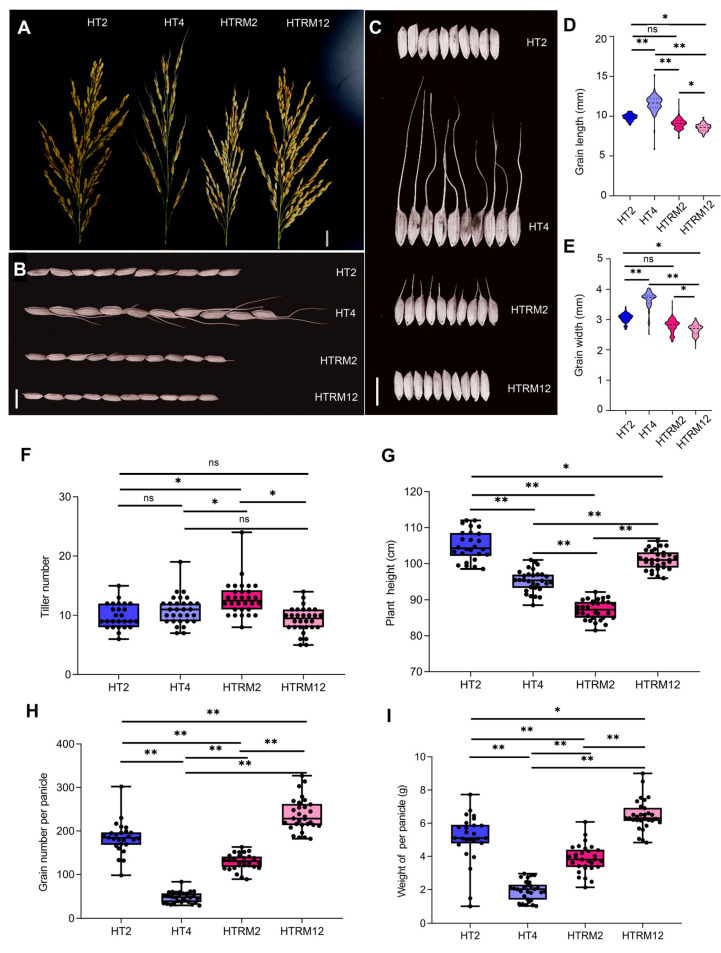
Agronomic traits of HT2, HT4, HTRM2, and HTRM12. (**A**) Comparison of spike shape of each sample. Bar, 1 cm; (**B**) comparison of seed length of each sample. Bar, 1 cm; (**C**) comparison of seed width of each sample. Bar, 1 cm; (**D**) grain length of each sample; (**E**) grain width of each sample; (**F**) tiller number of each sample; (**G**) plant height of each sample; (**H**) grain number per panicle of each sample; (**I**) weight of per panicle of each sample. The values in (**D**,**E**) show the mean ± SD (*n* > 300 seeds). The values in (**F**–**I**) show the mean ± SD (*n* = 30 plants). Significant differences: ns, no significance; *, *p* < 0.05; **, *p* < 0.01 (Student’s *t*-test).

**Figure 3 ijms-25-09012-f003:**
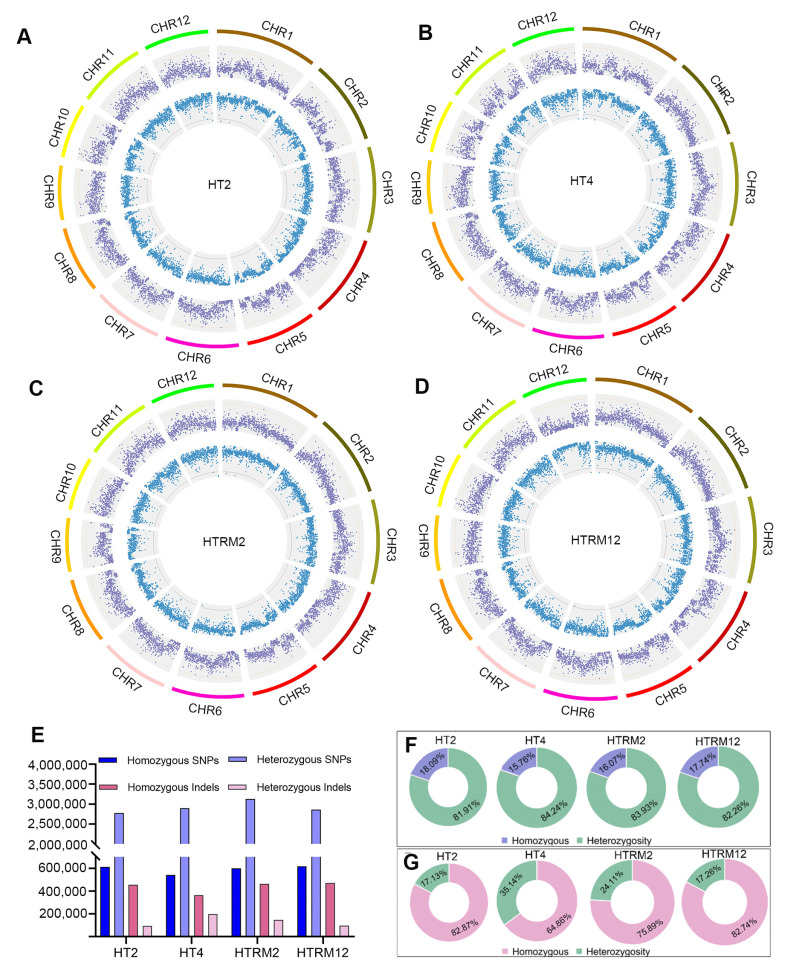
Number statistics and distribution of SNPs and InDels on chromosomes for HT2, HT4, HTRM2, and HTRM12. (**A**) Distribution of genomic variants in HT2 on chromosomes; (**B**) distribution of genomic variants in HT4 on chromosomes; (**C**) distribution of genomic variants in HTRM2 on chromosomes; (**D**) distribution of genomic variants in HTRM12 on chromosomes. Circle 1: chromosomes; Circle 2: purple loci indicate the distribution of genomic SNP densities; Circle 3: blue loci indicate the distribution of genomic InDel densities. Circle 1: chromosomes; (**E**) statistics of homozygous SNP, homozygous InDel, total SNP, and total InDel of each sample; (**F**) proportion of homozygous SNP and heterozygous SNP in each sample; (**G**) proportion of homozygous InDel and heterozygous InDel in each sample.

**Figure 4 ijms-25-09012-f004:**
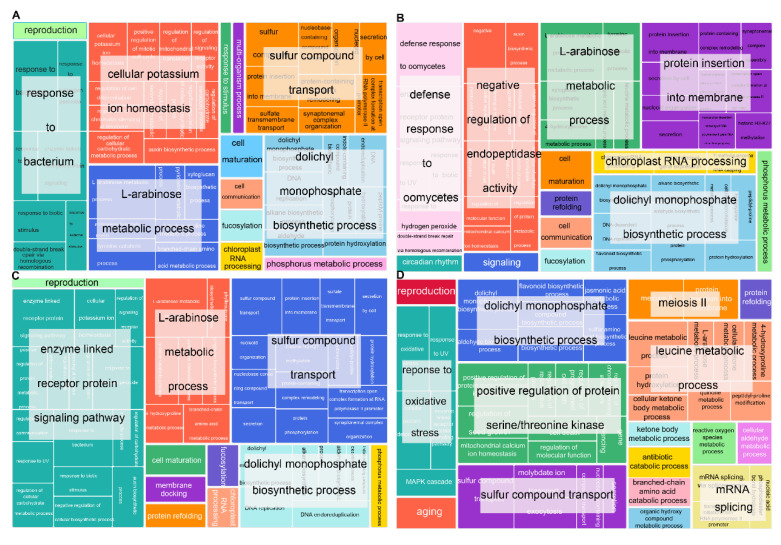
GO terms enrichment analysis results of genes with homozygous SNP mutations in HT2, HT4, HTRM2, and HTRM12. (**A**) GO terms enrichment analysis results of genes with homozygous SNP mutations in HT2; (**B**) GO terms enrichment analysis results of genes with homozygous SNP mutations in HT4; (**C**) GO terms enrichment analysis results of genes with homozygous SNP mutations in HTRM2; (**D**) GO terms enrichment analysis results of genes with homozygous SNP mutations in HTRM12. The results were visualized by the “TreeMap” view of REVIGO. Each rectangle is a single cluster representative. The representatives are joined into “superclusters” of loosely related terms, visualized with different colors. The size of the rectangles is adjusted to reflect the *p*-value.

**Figure 5 ijms-25-09012-f005:**
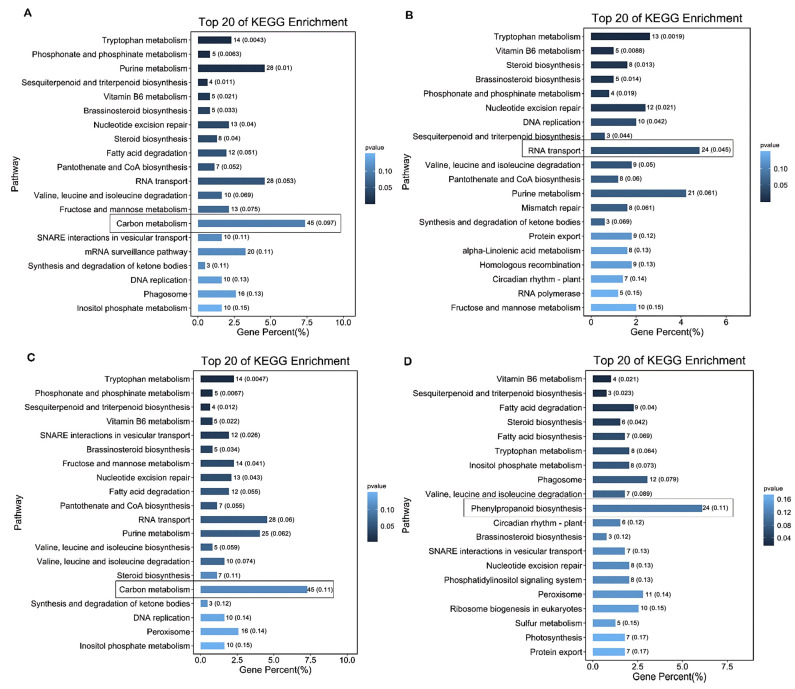
KEGG enrichment analysis results of genes with homozygous SNP mutations in HT2, HT4, HTRM2, and HTRM12. (**A**) Top 20 of KEGG enrichment analysis results of genes with homozygous SNP mutations in HT2; (**B**) top 20 of KEGG enrichment analysis results of genes with homozygous SNP mutations in HT4; (**C**) top 20 of KEGG enrichment analysis results of genes with homozygous SNP mutations in HTRM2; (**D**) top 20 of KEGG enrichment analysis results of genes with homozygous SNP mutations in HTRM12. The darker the blue, the smaller the *p*-value. The labeled boxes indicate the entries with the highest number of genes enriched in the KEGG metabolic pathway for each of these four materials.

**Figure 6 ijms-25-09012-f006:**
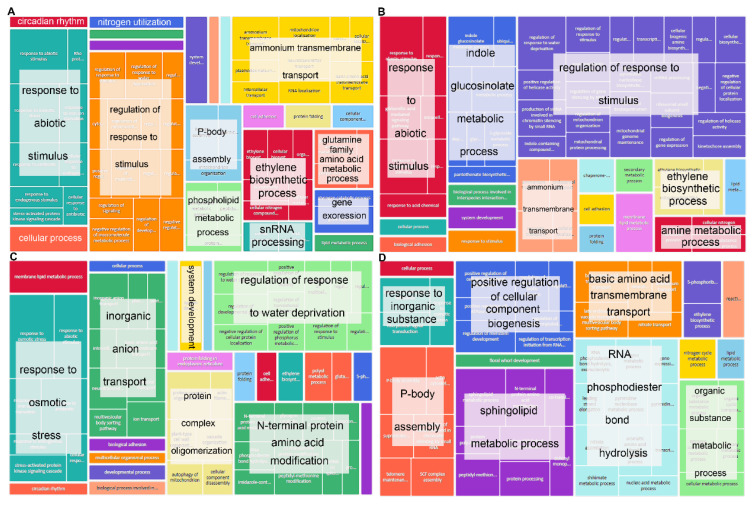
GO terms enrichment analysis results of genes with homozygous InDel mutations in HT2, HT4, HTRM2, and HTRM12. (**A**) GO terms enrichment analysis results of genes with homozygous InDel mutations in HT2; (**B**) GO terms enrichment analysis results of genes with homozygous InDel mutations in HT4; (**C**) GO terms enrichment analysis results of genes with homozygous InDel mutations in HTRM2; (**D**) GO terms enrichment analysis results of genes with homozygous InDel mutations in HTRM12. The results were visualized by the “TreeMap” view of REVIGO. Each rectangle is a single cluster representative. The representatives are joined into “superclusters” of loosely related terms, visualized with different colors. The size of the rectangles is adjusted to reflect the *p*-value.

**Figure 7 ijms-25-09012-f007:**
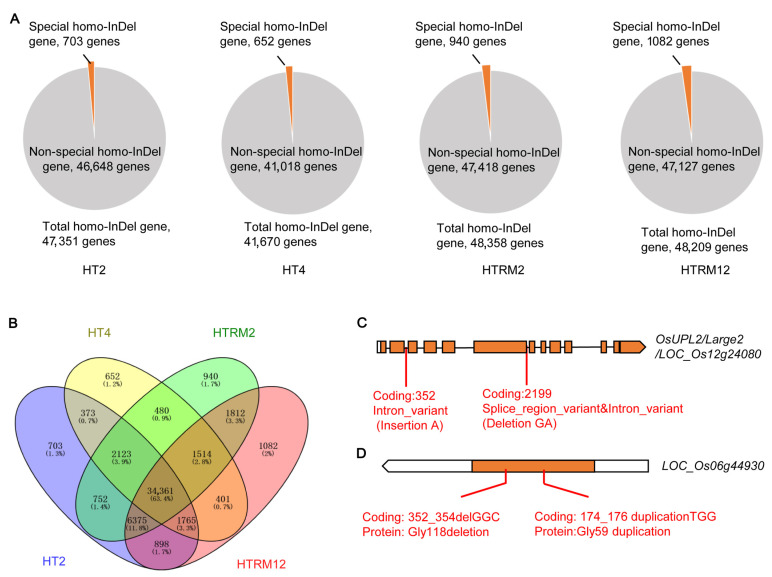
The awn genes were identified by GWAS phenotype enrichment analysis for special homo-InDel gene sets of four materials. (**A**) Statistics of homo-InDel genes and special homo-InDel genes for four materials; (**B**) the Venn diagram of homo-InDel genes of four materials; (**C**,**D**) the InDel variations in coding regions of awn genes identified by GWAS phenotype enrichment analysis (see Appendix A). Rectangles represent exons, orange rectangles represent protein-coding regions, and lines represent introns.

**Figure 8 ijms-25-09012-f008:**
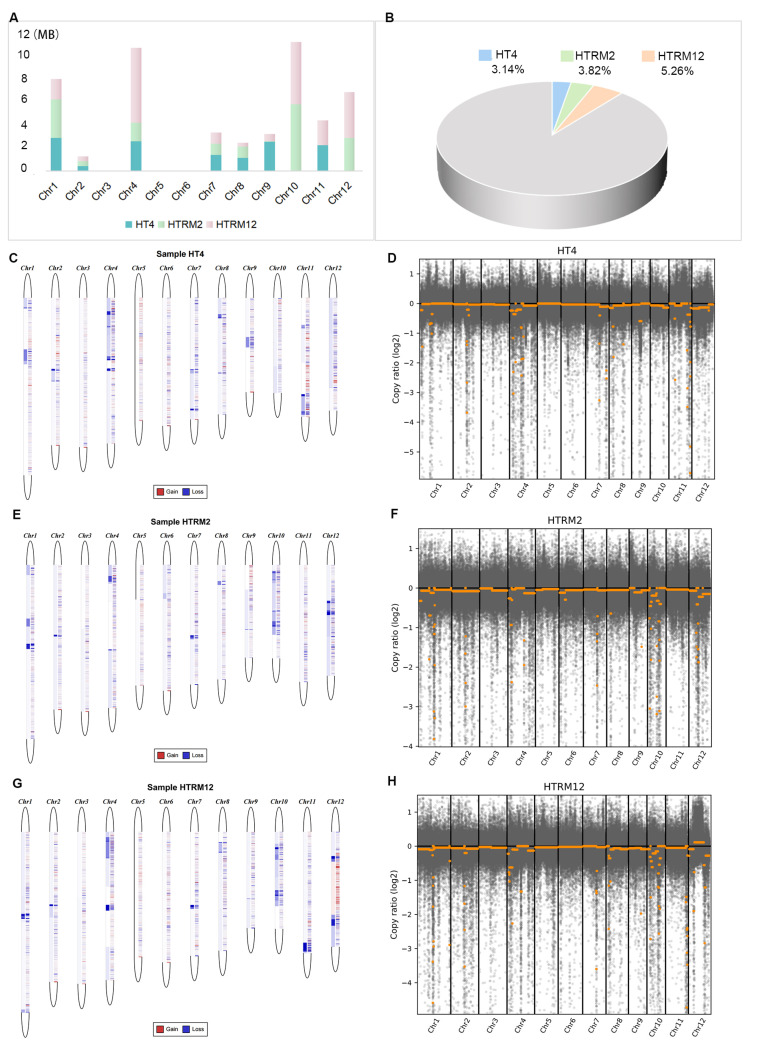
Identification and comparison of genomic CNV intervals of HT4, HTRM2, and HTRM12. (**A**) Distribution and size of the CNVs of HT4, HTRM2, and HTRM12 on the chromosome; the horizontal coordinates indicate the different chromosomes, and the vertical coordinates indicate the size of the CNV. Different colors indicate different samples; (**B**) the proportion of the specific CNV interval occupied by each sample; (**C**,**D**) distribution of differential CNVs on different chromosomes in HT4 relative to HT2; (**E**,**F**) distribution of differential CNVs on different chromosomes in HTRM2 relative to HT2; (**G**,**H**) distribution of differential CNVs on different chromosomes in HTRM12 relative to HT2; in (**C**,**E**,**G**) Red, increased copy number; blue, decreased copy number; left, increase or decrease in CNS (Copy Number Segment); right, increase and decrease in CNR (Copy Number Region); in (**D**,**F**,**H**) gray dot, CNR; yellow dot, CNS.

**Figure 9 ijms-25-09012-f009:**
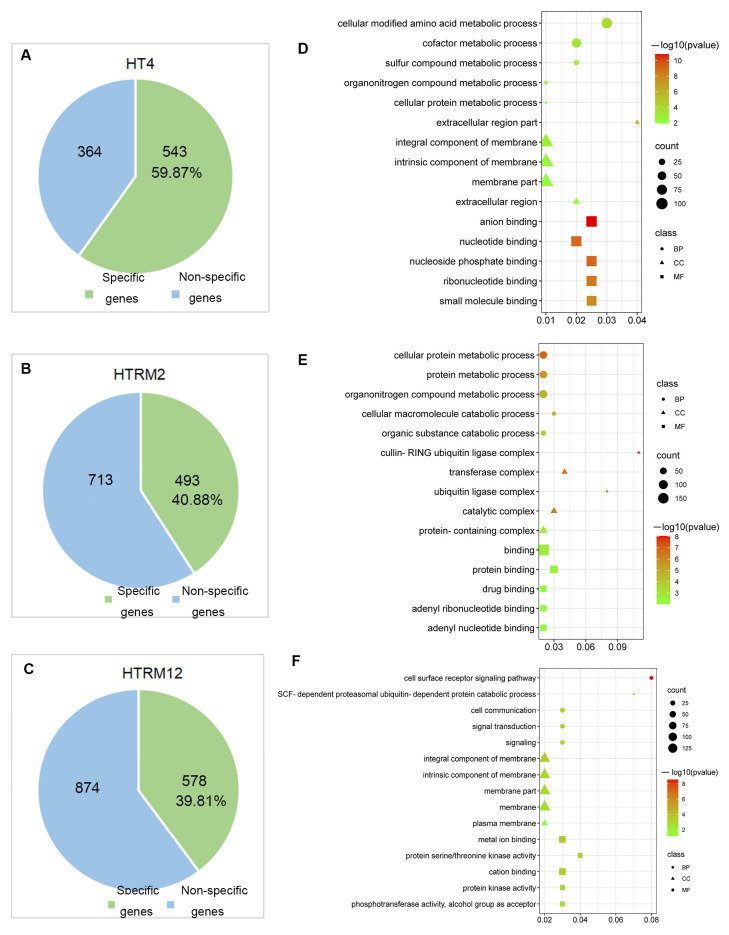
GO Terms enrichment analysis of genes within the specific CNV intervals of HT4, HTRM2, and HTRM12. (**A**–**C**) The proportion of specific genes for HT4, HTRM2, and HTRM12; (**D**–**F**) top 5 GO terms in each BP, CC, and MF enrichment analysis of genes within the specific CNV intervals of HT4, HTRM2, and HTRM12. Circles represent biological processes, triangles represent cellular components, and squares represent molecular functions; the size of the graphs indicates the number of genes.

**Table 1 ijms-25-09012-t001:** The disease and pest resistance test of hybrid variety HTRM12.

Variety	Susceptibility Index of Rice Blast	Susceptibility Index of Bacterial Leaf Blight	Susceptibility Index of Brown Planthopper
HTRM12	5.2	5	9
FLY4 (CK)	6.5	5	9

Note: The numbers of the index range from 0 to 9 grade. A smaller number means more resistance, and a larger number means more susceptible (for the test method, see Section 4).

## Data Availability

All data generated during the current study are available in online repositories and the Appendix A. The genome sequencing data can be found in the National Center for Biotechnology Information (NCBI) Short Read Archive under accession number PRJNA1017988.

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
