# Peer review of "Genome Resequencing for Autotetraploid Rice and Its Closest Relatives Reveals Abundant Variation and High Potential in Rice Breeding"

_ijms, 2024, doi:10.3390/ijms25169012_

Round 1

Reviewer 1 Report

Comments and Suggestions for Authors

Comments:

The manuscript by Yachun Zhang et al., titled: Genome Resequencing for Autotetraploid Rice and Its Closest Relatives Reveals Abundant Variation and High Potential in Rice Breeding, presents the feasibility that polyploid rice could be used as mutation carrier for creating variations. Although the authors have done significant findings to justify their claim, the manuscript still needs significant improvement before it can be considered for publication.

1.    The abstract should be clearer in highlighting the novelty and main findings of the study.

2.    Provide a more detailed background on the significance of autotetraploid rice and its potential in breeding. This will help readers understand the context and importance of the study.

3.    Include more details on the breeding process of HTRM12. Specify the conditions under which the field trials were conducted.

4.    Add a section on the statistical methods used for data analysis.

5.    While the results section provides a comparative analysis of the agronomic traits between the hybrid rice variety HTRM12 and the control FLY4, it lacks detailed genetic analysis explaining the observed differences. Including a thorough examination of specific genetic loci or markers associated with these traits would significantly enhance the depth of the study. For example, incorporating data from quantitative trait loci (QTL) mapping or genome-wide association studies (GWAS) could provide insights into the genetic basis of the observed phenotypic variations.

6.    The authors should elaborate on the implications of their findings for future rice breeding programs and genetic research in the discussion.

Comments on the Quality of English Language

Comments:

The manuscript by Yachun Zhang et al., titled: Genome Resequencing for Autotetraploid Rice and Its Closest Relatives Reveals Abundant Variation and High Potential in Rice Breeding, presents the feasibility that polyploid rice could be used as mutation carrier for creating variations. Although the authors have done significant findings to justify their claim, the manuscript still needs significant improvement before it can be considered for publication.

1.    The abstract should be clearer in highlighting the novelty and main findings of the study.

2.    Provide a more detailed background on the significance of autotetraploid rice and its potential in breeding. This will help readers understand the context and importance of the study.

3.    Include more details on the breeding process of HTRM12. Specify the conditions under which the field trials were conducted.

4.    Add a section on the statistical methods used for data analysis.

5.    While the results section provides a comparative analysis of the agronomic traits between the hybrid rice variety HTRM12 and the control FLY4, it lacks detailed genetic analysis explaining the observed differences. Including a thorough examination of specific genetic loci or markers associated with these traits would significantly enhance the depth of the study. For example, incorporating data from quantitative trait loci (QTL) mapping or genome-wide association studies (GWAS) could provide insights into the genetic basis of the observed phenotypic variations.

6.    The authors should elaborate on the implications of their findings for future rice breeding programs and genetic research in the discussion.

Author Response

Dear Editor and Reviewer 1:

Thank you very much for taking the time to review this manuscript entitled “Genome Resequencing for Autotetraploid Rice and Its Closest Relatives Reveals Abundant Variation and High Potential in Rice Breeding” (Manuscript ID: ijms-3116800). Those comments are all valuable and very helpful for revising and improving our paper. We have studied comments carefully and have made revisions which we hope meet with approval. Please find the detailed responses below and the corresponding revisions/corrections highlighted in blue in the re-submitted files.

Reviewer 1

Comments and Suggestions for Authors

Comments:

The manuscript by Yachun Zhang et al., titled: Genome Resequencing for Autotetraploid Rice and Its Closest Relatives Reveals Abundant Variation and High Potential in Rice Breeding, presents the feasibility that polyploid rice could be used as mutation carrier for creating variations. Although the authors have done significant findings to justify their claim, the manuscript still needs significant improvement before it can be considered for publication.

1.The abstract should be clearer in highlighting the novelty and main findings of the study.

Response: Thank you for your suggestion, the novelty and main findings of the study are as follows:

  • Use the tetraploid rice HT4 as a mutant carrier to breed a new excellent diploid material HTRM2, and then bred the hybrid rice variety HTRM12, innovatively. It provides a new way of application for tetraploid rice breeding;
  • Analyze the important agronomic traits and genome-wide variations of those closest relatives, Haitian diploid (HT2), Haitian tetraploid (HT4), HTRM2, and HTRM12 in detail, systematically, and find that HT4 and HTRM2 had abundant phenotypic and genomic variations compared to HT2. HTRM2 can inherit important traits and variations from HT4. It provides the genomic information and new breeding materials for tetraploid rice breeding.

To address this question, we have modified the abstract section in blue highlight.

2.Provide a more detailed background on the significance of autotetraploid rice and its potential in breeding. This will help readers understand the context and importance of the study.

Response: Thank you for your suggestion. We have added the details on the significance of autotetraploid rice and its potential in breeding in background section. Please see corresponding position in blue highlight in text.

3.Include more details on the breeding process of HTRM12. Specify the conditions under which the field trials were conducted.

Response: Thank you for your suggestion. We have added the details on the breeding process of HTRM12 and the field trial condition in the “4.1 Rice material and phenotype examination” part in the “Materials and Methods” section in blue highlight in text.

4.Add a section on the statistical methods used for data analysis.

Response: Thank you for your suggestion. We have added a paragraph “4.9 Statistical methods” in the “Materials and Methods” section in blue highlight in text.

5.While the results section provides a comparative analysis of the agronomic traits between the hybrid rice variety HTRM12 and the control FLY4, it lacks detailed genetic analysis explaining the observed differences. Including a thorough examination of specific genetic loci or markers associated with these traits would significantly enhance the depth of the study. For example, incorporating data from quantitative trait loci (QTL) mapping or genome-wide association studies (GWAS) could provide insights into the genetic basis of the observed phenotypic variations.

Response: Thank you for your good suggestion. The rice variety FLY4 is a commonly used for control variety (CK) in the commercial variety test in the middle and lower reaches of the Yangtze River in China. We compared the agronomic traits of HTRM12 to that of FLY4 to show that HTRM12 performed excellent in the middle and lower reaches of the Yangtze River. The analysis of genetic differences between HTRM12 and FLY4 is not the main purpose of this article, and the genomic data of FLY4 is unavailable for us now.

6.The authors should elaborate on the implications of their findings for future rice breeding programs and genetic research in the discussion.

Response: Thank you for your suggestion. We rewrote the discussion section. Please see the responding section in blue highlight in the text.

Comments on the Quality of English Language

Reviewer 2 Report

Comments and Suggestions for Authors

The authors have bred a new two-line hybrid rice variety (HTRM12) and performed detailed comparative analysis with close relatives using multiple phenotypic investigations, genome resequencing, and bioinformatics analysis. The results are interesting and useful for future rice breeding programs. For example, results suggest that HTRM2 can inherit important traits and variations from HT4 suggesting that tetraploid self-reverted diploid has high potential in creating excellent breeding materials. The study provides insights into parental utilization of tetraploid rice and new insights into rice breeding. There are a few minor suggestions for improvement.

1.       Line 7. Y Liang You Duo Hui 14 (HTRM12) In full form you mentioned 14 but in abbreviation 12?

2.       The last sentence of the abstract needs to be revised for clarity of meaning. Split the sentence into 2 sentences and the same suggestion for lines 102-105.

3.       Some figures can be shifted to the supplementary data ( i.e. figures 4 and 6).

4.       It will be better to show figures 4 and 6 like other Go terms shown in figure 9.

5.       Lines 506 to 523 are repetitions of lines mentioned in the introduction and results section. Most of the lines need to be deleted and the remaining need to be rewritten for clarity of meaning.

6.       Line 108 no need to write Hai Tian as you already mentioned in line 96.

7.       The discussion part needs to be revised. Here you have again mentioned the results. Support your results with strong scientific evidence and logic.

Comments on the Quality of English Language

Minor spell check is required

Author Response

Dear Editor and Reviewer 2:

Thank you very much for taking the time to review this manuscript entitled “Genome Resequencing for Autotetraploid Rice and Its Closest Relatives Reveals Abundant Variation and High Potential in Rice Breeding” (Manuscript ID: ijms-3116800). Those comments are all valuable and very helpful for revising and improving our paper. We have studied comments carefully and have made revisions which we hope meet with approval. Please find the detailed responses below and the corresponding revisions/corrections highlighted in blue in the re-submitted files.

Reviewer 2

Comments and Suggestions for Authors

The authors have bred a new two-line hybrid rice variety (HTRM12) and performed detailed comparative analysis with close relatives using multiple phenotypic investigations, genome resequencing, and bioinformatics analysis. The results are interesting and useful for future rice breeding programs. For example, results suggest that HTRM2 can inherit important traits and variations from HT4 suggesting that tetraploid self-reverted diploid has high potential in creating excellent breeding materials. The study provides insights into parental utilization of tetraploid rice and new insights into rice breeding. There are a few minor suggestions for improvement.

1.Line 7. Y Liang You Duo Hui 14 (HTRM12) In full form you mentioned 14 but in abbreviation 12?

Response: Thank you. “HTRM12” is a sample ID used in the genome sequencing experiment, there is no special meaning for this symbol, and this symbol is continued to be used in order to avoid confusion when we prepared the article.

2.The last sentence of the abstract needs to be revised for clarity of meaning. Split the sentence into 2 sentences and the same suggestion for lines 102-105.

Response: Thank you for your suggestion. We revised the last sentence of the abstract.

3.Some figures can be shifted to the supplementary data (i.e. figures 4 and 6).

Response: Thank you for your suggestion.

The Figure 4 and Figure 6 show the results of GO terms enrichment analysis of genes with homozygous SNPs and homozygous InDels in the “TreeMap” view of REVIGO style for the four materials, respectively. These results can easily reflect the specific enrichment GO terms for the homologous variation genes in each material, respectively, which is helpful to understand the characteristics of each material. It clearly shows which items are shared by different materials and which items are more specific for specific material. For example, we can easily find that the sharing items related to stress and environmental responses in HT2, HT4, and HTRM2 in Figure 6 A, B, C. These results are an important part of genomic data analysis. Therefore, we think they should be put in the main text.

REVIGO is a Web server that summarizes long, unintelligible lists of GO terms by finding a representative subset of the terms using a simple clustering algorithm that relies on semantic similarity measures. Furthermore, REVIGO visualizes this non-redundant GO term set in multiple ways to assist in interpretation (See reference: Supek F, Bosnjak M, Skunca N, Smuc T (2011) REVIGO Summarizes and Visualizes Long Lists of Gene Ontology Terms. PLoS ONE 6(7): e21800. doi:10.1371/journal. pone. 0021800). So, compared with the traditional GO enrichment presentation (such as Figure 9), the REVIGO has more advantages in the long, unintelligible lists of GO terms.

4.It will be better to show figures 4 and 6 like other Go terms shown in figure9.

Response: Thank you for your suggestion. See the above response.

5.Lines 506 to 523 are repetitions of lines mentioned in the introduction and results section. Most of the lines need to be deleted and the remaining need to be rewritten for clarity of meaning.

Response: Thank you for your suggestion. We rewrote the discussion section. Please see the responding section in blue highlight in the text.

6.Line 108 no need to write Hai Tian as you already mentioned in line 96.

Response: Thank you for reminding us. We have removed the "Hai Tian" in the corresponding position, see the corresponding position in blue highlight.

7.The discussion part needs to be revised. Here you have again mentioned the results. Support your results with strong scientific evidence and logic.

Response: Thank you for your suggestion again. We rewrote the discussion section. Please see the corresponding section in blue highlight in the text.

Comments on the Quality of English Language

Minor spell check is required

Reviewer 3 Report

Comments and Suggestions for Authors

The manuscript entitled "Genome Resequencing for Autotetraploid Rice and Its Closest Relatives Reveals Abundant Variation and High Potential in Rice Breeding" aims to "provides insights into parental utilization of tetraploid 102 rice, gene interval traceability of important traits, and pedigree analysis of new varieties 103 in rice breeding for the future and also provides an important reference for studying the 104 genetic composition of important backbone germplasm at the genome level." I have the following comments:

- The manuscript is quite complex and has a variety of methods involved to characterize the new line developed. I congratulate the authors on that.  Yet, the discussion is very poor. Without further exploration of the results found, the manuscript is quite technical or descriptive. 

- Material and methods lack many specific details, preventing the replication of this study:

1- "HT2, HT4, HTRM2 and 590 HTRM12 were grown under field condition following regular field management in Hai- 591 nan province of China in the winter of 2021 year. These lines need to be described, as well as field conditions.

2- A description of the morphological traits measured is needed.

3- Statistical analysis is very poorly presented. A t-student is mentioned but in order to do this, the normality of data should have been checked previously.

4- DNA was extracted using CTAB but without further details.

5- The same goes for sequencing, as well as the information programs used. The conditions should be mentioned since the results are highly correlated with the factors used in these programs. 

6- Replicates were used in this study?

8- Library sequencing has no details of methods or even the type of instrument used.

9- No data about quality control and results have been included. 

10- Which genome was used to obtain SNPs and the subsequent data?

11- As authors are dealing with polyploid and diploids, specific methods should have been employed to overcome the duplication of reads, genes and so on. Nothing is said about that.

12- Several results are stated in SI files but without a clear connection with M&Ms.

13- Data availability: data of PRJNA1017988 was not found in the repository.

Comments on the Quality of English Language

See above.

Author Response

Dear Editor and Reviewer 3:

Thank you very much for taking the time to review this manuscript entitled “Genome Resequencing for Autotetraploid Rice and Its Closest Relatives Reveals Abundant Variation and High Potential in Rice Breeding” (Manuscript ID: ijms-3116800). Those comments are all valuable and very helpful for revising and improving our paper. We have studied comments carefully and have made revisions which we hope meet with approval. Please find the detailed responses below and the corresponding revisions/corrections highlighted in blue in the re-submitted files.

Reviewer 3

Comments and Suggestions for Authors

The manuscript entitled "Genome Resequencing for Autotetraploid Rice and Its Closest Relatives Reveals Abundant Variation and High Potential in Rice Breeding" aims to "provides insights into parental utilization of tetraploid rice, gene interval traceability of important traits, and pedigree analysis of new varieties in rice breeding for the future and also provides an important reference for studying the genetic composition of important backbone germplasm at the genome level." I have the following comments:

- The manuscript is quite complex and has a variety of methods involved to characterize the new line developed. I congratulate the authors on that. Yet, the discussion is very poor. Without further exploration of the results found, the manuscript is quite technical or descriptive.

Response: Thanks for your comments.

  At present, the breeders are promoting the application of tetraploid rice by cultivating highly fertile tetraploid materials and constructing tetraploid heterosis utilization systems and have made some progress (See Introduction section). However, there is no commercial variety directly bred with tetraploid. In this study, we used tetraploid rice HT4 as a mutant carrier to breed a new excellent diploid material HTRM2, and then bred the hybrid rice variety HTRM12. Furthermore, we comparatively analyzed the important agronomic traits and genome-wide variations of those closest relatives, Haitian diploid (HT2), Haitian tetraploid (HT4), HTRM2, and HTRM12 in detail, based on multiple phenotypic investigation, genome resequencing and bioinformatics analysis. Our results indicate that genomic variation can occur in the process of chromosome ploidy change of rice, and some variations can be inherited to the reverted diploid from tetraploid rice. The reverted diploid rice has a large variation relative to the original diploid rice, so it can be used to breed high-yield and strong heterosis rice materials. This study provides the genomic information, new breeding materials and new way of application for tetraploid rice breeding.

To improve the quality of discussion section of the manuscript, we rewrote the discussion section.

- Material and methods lack many specific details, preventing the replication of this study:

1- "HT2, HT4, HTRM2 and 590 HTRM12 were grown under field condition following regular field management in Hai- 591 nan province of China in the winter of 2021 year. These lines need to be described, as well as field conditions.

Response: Thank you for your suggestion. We have added the more details on the rice materials and the field trial conditions in “4.1 Rice material and phenotype examination” in the “Materials and Methods” section in blue highlight in text.

2- A description of the morphological traits measured is needed.

Response: Thank you for your suggestion. See the above response.

3- Statistical analysis is very poorly presented. A t-student is mentioned but in order to do this, the normality of data should have been checked previously.

Response: Thanks for your suggestion. We conducted a normality test on the data, and wrote a separate paragraph of the data statistical method “4.9 Statistical methods” based on the data test results in the “Materials and Methods”section. Additionally, we also revised the corresponding figure legend of Figure1, Figure 2 and the note of Table 1 in blue highlight.

4- DNA was extracted using CTAB but without further details.

Response: Thanks for your suggestion. We have revised the “4.4 Genome sequencing and data quality assessment” part in the “Materials and Methods”section, and added more details on the genome sequencing and data quality assessment. The important parameters of the experiment, the important parameters of the data analysis and the software version used in this study were added in this part. Please see the corresponding section in blue highlight in the text.

5- The same goes for sequencing, as well as the information programs used. The conditions should be mentioned since the results are highly correlated with the factors used in these programs.

Response: Thank you for your suggestion. See the above response.

6- Replicates were used in this study?

Response: Thank you. Yes, all of our field trials were repeated, more details can be found in “4.1 Rice material and phenotype examination” part.

7- Library sequencing has no details of methods or even the type of instrument used.

Response: Thanks for your suggestion. We revised the “4.4 Genome sequencing and data quality assessment” part in the “Materials and Methods”section, and added more details on the genome sequencing and data quality assessment. please see the corresponding section in blue highlight in the text.

8- No data about quality control and results have been included.

Response: Thanks for your suggestion. We have performed quality control on the genome sequencing data, and the corresponding results can be found in the supplementary figures and tables, which linked in the corresponding positions in the “Materials and Methods” section in the new vision of the manuscript.

To address the question, we revised the “4.4 Genome sequencing and data quality assessment” part in the “Materials and Methods”section, and added more details on the genome sequencing and data quality assessment. please see the corresponding section in blue highlight in the text.

9- Which genome was used to obtain SNPs and the subsequent data?

Response: Thanks. In this study, the Nipponbare genome (MSU-release7) was used for SNPs and InDels detection. We have added the information in the “4.4 Genome sequencing and data quality assessment” part in the “Materials and Methods”section in our new version of the manuscript.

10- As authors are dealing with polyploid and diploids, specific methods should have been employed to overcome the duplication of reads, genes and so on. Nothing is said about that.

Response: Thanks for your suggestion. In this study, four materials (except for autotetraploid HT4, the other three materials are diploid rice) were subjected to parallel library construction, sequencing and mutation detection. We mainly analyzed the SNP and InDel variations in this study. The SNP and InDel variations are actually DNA base variations, and the homozygous heterozygosity is detected according to the number of reads of different haplotypes, so the chromosome ploidy has little effect on the detection of SNP and InDel variations. Moreover, most of researchers use the similar method to detect SNP and InDel variations and do not consider the “polyploid problem” when studing the SNP and InDel variations of polyploid rice in publication (See the reference 21,22,27,28,34,35 in Introduction section of the manuscript).

We used CNVkit software for CNV calling, which is based on sequence coverage by short reads (read depth method) (See reference: Talevich E, Shain AH, Botton T, Bastian BC,2016, CNVkit: Genome-Wide Copy Number Detection and Visualization from Targeted DNA Sequencing. PLoS Comput Biol 12(4): e1004873. doi:10.1371/journal.pcbi.1004873). Although the ploidy of the four materials is different, the genome re-sequencing data capacity is similar among them. In order to simplify the analysis of CNVs, we did not employ any specific methods to overcome the duplication of reads, genes and so on. Because that we think this bias may be eliminated, to a large extent, between chromosome ploidy and re-sequencing data capacity.

11- Several results are stated in SI files but without a clear connection with M&Ms.

Response: Thanks for your reminding. In the new version of manuscript, we have checked all SI files linked in the text. We have linked the SI files about the results of genome sequencing quality control, for example, Figure S1, Figure S2, Figure S3, Table S4, Table S5 to “4.4 Genome sequencing and data quality assessment” part in the “Materials and Methods”section. The corresponding part in “Materials and Methods” section were modified in blue highlight.

12- Data availability: data of PRJNA1017988 was not found in the repository.

Response: Thanks for your reminding. Probably due to the problem of repository setting in NCBI, the data can not be accessed normally before. We have checked it and the data of PRJNA1017988 can be accessed now.

Comments on the Quality of English Language

See above.

Round 2

Reviewer 3 Report

Comments and Suggestions for Authors

The authors have addressed all queries. I do not agree with the methods used to track genomic differences in polyploids, but this is up to the authors. 

Comments on the Quality of English Language

The authors have improved the text but there are still typos and grammatical mistakes. I leave this to the editorial team.